# Structural basis for the adaptation and function of chlorophyll *f* in photosystem I

Koji Kato[1,14], Toshiyuki Shinoda[2,14], Ryo Nagao[1,14], Seiji Akimoto[3], Takehiro Suzuki[4], Naoshi Dohmae[4], Min Chen[5], Suleyman I. Allakhverdiev[6,7,8,9,10], Jian-Ren Shen[1*], Fusamichi Akita[1,11*], Naoyuki Miyazaki[12,13*] & Tatsuya Tomo[2*]

Chlorophylls (Chl) play pivotal roles in energy capture, transfer and charge separation in photosynthesis. Among Chls functioning in oxygenic photosynthesis, Chl *f* is the most red-shifted type first found in a cyanobacterium *Halomicronema hongdechloris*. The location and function of Chl *f* in photosystems are not clear. Here we analyzed the high-resolution structures of photosystem I (PSI) core from *H. hongdechloris* grown under white or far-red light by cryo-electron microscopy. The structure showed that, far-red PSI binds 83 Chl *a* and 7 Chl *f*, and Chl *f* are associated at the periphery of PSI but not in the electron transfer chain. The appearance of Chl *f* is well correlated with the expression of PSI genes induced under far-red light. These results indicate that Chl *f* functions to harvest the far-red light and enhance uphill energy transfer, and changes in the gene sequences are essential for the binding of Chl *f*.

[1] Research Institute for Interdisciplinary Science and Graduate School of Natural Science and Technology, Okayama University, Okayama 700-8530, Japan. [2] Faculty of Science, Tokyo University of Science, Tokyo 162-8601, Japan. [3] Graduate School of Science, Kobe University, Kobe 657-8501, Japan. [4] Biomolecular Characterization Unit, RIKEN Center for Sustainable Resource Science, Wako, Japan. [5] School of Life and Environmental Sciences, University of Sydney, Sydney, Australia. [6] K.A. Timiryazev Institute of Plant Physiology RAS, Moscow, Russia. [7] Institute of Basic Biological Problems RAS, Pushchino, Moscow Region, Russia. [8] M.V. Lomonosov Moscow State University, Moscow, Russia. [9] Moscow Institute of Physics and Technology, Dolgoprudny, Moscow region, Russia. [10] Institute of Molecular Biology and Biotechnology ANAS, -Baku, Azerbaijan. [11] Japan Science and Technology Agency, PRESTO, Saitama 332-0012, Japan. [12] Institute for Protein Research, Laboratory of Protein Synthesis and Expression, Osaka University, Osaka 565-0871, Japan. [13] Life Science Center for Survival Dynamics, Tsukuba Advanced Research Alliance, University of Tsukuba, Ibaraki 305-8577, Japan. [14] These authors contributed equally: Koji Kato, Toshiyuki Shinoda, Ryo Nagao. *email: shen@okayama-u.ac.jp; fusamichi_a@okayama-u.ac.jp; naomiyazaki@tara.tsukuba.ac.jp; tomo@rs.tus.ac.jp

Chlorophylls (Chl) play important roles such as light-harvesting, energy transfer, and electron transfer in photosynthetic organisms. In order to utilize various spectral components of the solar energy efficiently, oxyphototrophs have developed a variety of Chl species that differ in their molecular structures and absorption regions. All oxygenic photosynthesis organisms possess Chl a in both photosystem cores and light-harvesting antennas. Chls b and c are associated with the antennas but not the photosystem cores in different lineages of organisms. These Chl variations are important for adaptation of oxyphototrophs to various living environments, leading to the acquisition of every ecological niche[1]. In addition to these Chls, a largely red-shifted Chl, Chl f, was first found in a filamentous cyanobacterium Halomicronema hongdechloris[2], and subsequently in several other species[3–5]. The Chl species in the reaction center (RC) of photosystem I (PSI) whose structure has been reported from cyanobacteria to higher plants are all Chl a[6–15]. However, the location and exact function of Chl f in the photosystems are not clear, although some reports have suggested that one or multiple Chl f molecules may function in the electron transfer chain (ETC) of PSI[16–19]. The Chl f content in H. hongdechloris depends on the quality of the cultivation light, e.g., the Chl f content is negligible under white fluorescent light, but reaches to 8–12.5% of the total Chl under far-red light (>700 nm)[20–23], Chl f has the structure of [2-formyl]-Chl a, and its absorption maximum at the Qy band is ~40 nm red-shifted relative to those of Chl a in organic solutions. Some cyanobacteria can use far-red light to carry out oxygenic photosynthesis by accumulating Chl f through a process of far-red light photoacclimation (FaRLiP)[3,22–25]. During FaRLiP, cells synthesize Chl f and extensively remodel their photosynthetic apparatus by changing the expressions of core subunits of PSI, PSII, and the phycobilisome. This switch allows cyanobacteria to grow under far-red light, and the Chl f synthase has been found to be located in the FaRLiP gene cluster[23]. Studies on the distribution of Chl f-containing organisms in bacterial mat showed that they are distributed 4–6.5 mm below the surface where the intensity of far-red light is higher than photosynthetically active radiation[26].

Among the two photosystems, PSI catalyzes the light-driven electron transfer from the plastocyanin (or cytochrome $c_6$) at the lumenal side of the thylakoid membranes to ferredoxin at the stromal side. The PsaA and PsaB subunits bind Chl a/Chl a′ as the primary electron donor (P700), and $A_0$ (Chl), $A_1$ (quinone), and $F_x$ (4Fe-4S cluster) as the electron acceptors. $A_{-1}$ is a Chl located between P700 and $A_0$. Recent spectroscopic measurements have suggested that Chl f may be present at the $A_{-1}$ site in the PSI cores[16]. However, no structural evidence for the Chl f localization in PSI have been reported. Here we analyzed the high-resolution structures of the PSI core complexes from H. hongdechloris grown under white (white PSI) and far-red (far-red PSI) light conditions by cryo-electron microscopy (cryo-EM). We found no Chl f molecules in the white PSI but seven Chl f molecules in the far-red PSI. These Chl f are located in the periphery region of the PSI core but not in the ETC. Combining with our previous functional analyses[20,27,28], we conclude that Chl f functions in enhancing excitation energy transfer to the RC Chls rather than being involved in the ETC directly.

## Results

### Overall structures of white PSI and far-red PSI.
We purified the trimeric PSI cores from H. hongdechloris grown under white and far-red light conditions (Supplementary Fig. 1) and solved their structures by cryo-EM single particle analyses at resolutions of 2.35 and 2.41 Å, respectively (Supplementary Figs. 2–5 and Supplementary Table 1). Both PSI cores form homo-trimers

(Supplementary Fig. 4a, b), and the overall structures are similar to that of the cyanobacterial PSI trimer (containing only Chl a) reported previously[6,11], except for PsaF, PsaJ, and PsaX (and PsaK in the case of white PSI). Although the gene for PsaX was not found in the genome of H. hongdechloris, the genes for PsaF, PsaJ, and PsaK are present[29]. In the atomic model of the white PSI, each monomer contains five membrane-spanning subunits (PsaA, PsaB, PsaI, PsaL, and PsaM) and three stromal subunits (PsaC, PsaD, and PsaE) (Fig. 1a and Supplementary Fig. 6). The density map for PsaF, PsaJ, and PsaK of the white PSI are rather poor compared with that of the other assigned subunits, and hence, these three subunits were deleted in the structure (Supplementary Figs. 2g and 6). Sodium dodecyl sulfate polyacrylamide gel electrophoresis (SDS-PAGE) analysis of the purified PSI solution sample showed that it contained PsaF, PsaJ, and PsaK (Supplementary Fig. 1d and supplementary Table 2). Therefore, the poor densities for these three subunits in the cryo-EM map may be owing to the weak association of these subunits with PSI and partial dissociation of them during the grid preparation for cryo-EM.

In the atomic model of the far-red PSI, each monomer contains six membrane-spanning subunits (PsaA, PsaB, PsaI, PsaK, PsaL, and PsaM) and three stromal subunits (PsaC, PsaD, and PsaE) (Fig. 1b and Supplementary Fig. 3f). The densities for PsaF and PsaJ were practically not found even if we lowered the map contour to the noise level. As these two subunits were also detected in the solution sample by SDS-PAGE (Supplementary Fig. 1d), this result indicates an even weaker association of them with the PSI core. One of the reasons for this loose association of PsaF and PsaJ may be owing to the replacement of psaA, psaB, as well as psaF, psaI, psaL, and psaJ genes under the far-red light condition (see below)[3,29]. PsaF was previously suggested to be involved in binding of the electron donor, such as the c-type cytochrome[30]. We measured the light-induced difference absorption change of P700 for Thermosynechococcus elongatus PSI and far-red PSI, which showed that the difference absorption changes are almost the same between the two PSIs (Supplementary Fig. 7). This result supports the presence and functioning of PsaF in the purified far-red PSI core, and suggests that the absence of its density in the cryo-EM map may be caused by its loss during the grid preparation.

The root mean square deviation (RMSD) between the white and far-red PSI is 0.62 Å for 1855 $C_\alpha$ atoms from subunits whose structures were built in both PSI cores, suggesting a large similarity in the overall structures of the two PSI cores. However, it is known that far-red light induces an extensive remodeling of the photosynthetic apparatus of H. hongdechloris by changing the expression of genes encoding the PSI core subunits as well as the synthesis of pigments (including change of Chl a to Chl f) of PSI[22–25,29]. Indeed, the sequences of some subunits in the structure of the far-red PSI were found to be different from that of the white PSI. In our structure, PsaA_2, PsaB_2, PsaI_3, and PsaL_2 subunits in the white PSI were changed to PsaA_1, PsaB_1, PsaI_2, and PsaL_1 subunits in the far-red PSI, respectively. These four subunits are encoded by different genes with different amino-acid sequences between the white and far-red PSI, whereas the other subunits are encoded by the same genes[29]. The sequence identities for the four subunits, PsaA, PsaB, PsaI, and PsaL between the white and far-red PSI are 74.6%, 80.9%, 36.8%, and 43.4%, respectively, in this cyanobacterium. Multiple sequence alignment of these four subunits from three species of cyanobacteria that are known to undergo remodeling between white and far-red light conditions (H. hongdechloris, Leptolyngbya sp. strain JSC-1 and Chroococcidiopsis thermalis PCC7203) shows high homologies for each pair of the subunits grown under the same conditions (either white or far-red light

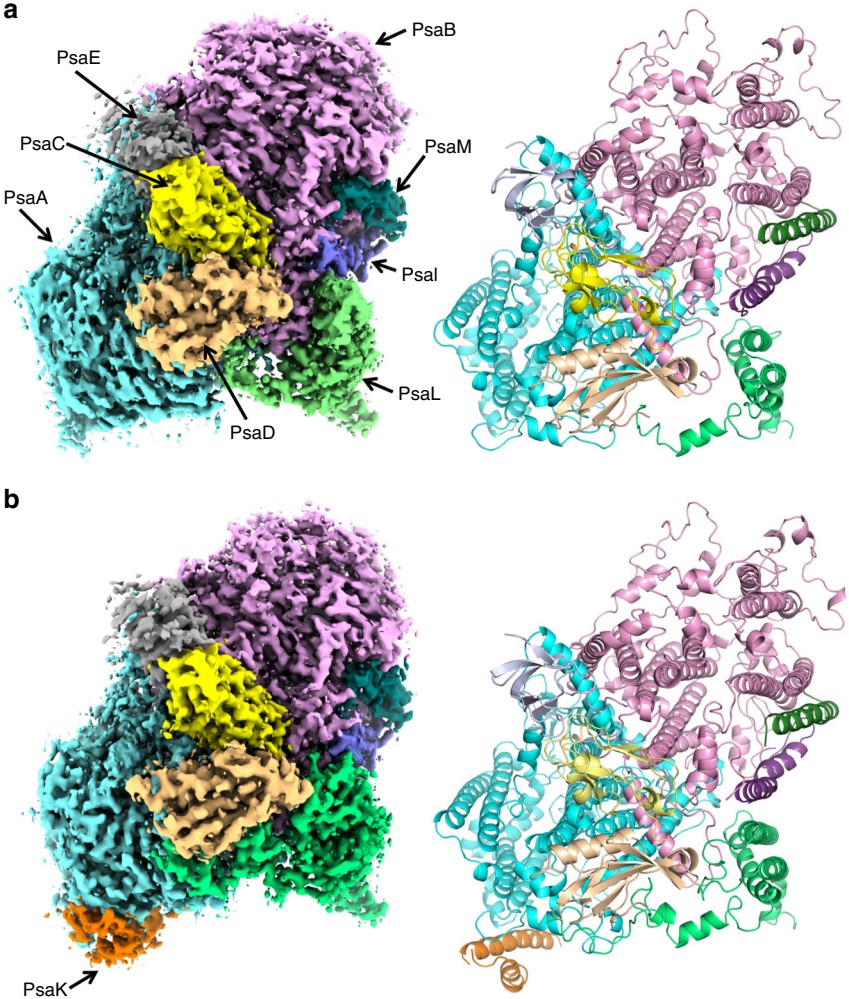

**Fig. 1 Structure of white PSI and far-red PSI monomers. a** Cryo-EM density map (left) and structure (right) of the white PSI monomer viewed along the membrane normal from the stromal side. **b** Cryo-EM density map (Left) and structure (Right) of the far-red PSI monomer viewed along the membrane normal from the stromal side.

conditions), but lower homologies between the same subunits expressed under white or far-red light conditions (Supplementary Figs. 8–11 and Supplementary Table 3). This is particular apparent for the PsaA and PsaI subunits. For example, the PsaA subunit has an average identity of 84.4% and 83.2% among cyanobacteria grown under far-red and white light conditions, respectively, whereas the average identity of PsaA between cyanobacteria grown under far-red and white light conditions is 76.2% (Supplementary Table 3). Furthermore, the PsaI subunit has an average identity of 65.0% and 51.8% among cyanobacteria grown under far-red and white light conditions, respectively, whereas the average identity between cyanobacteria grown under far-red and white light conditions is only 38.6%.

Several regions with different sequences between the subunits expressed under white and far-red light conditions were identified to be important for the structural differences between the white and far-red PSI. In the PsaA subunit, loop1 (Ala232-Asp244) located at the lumenal side of white PSI was deleted in the far-red PSI, whereas loop2 (Pro331-Asn338) is inserted in the far-red PSI (Supplementary Fig. 8), which was found to be located at the stromal side (Fig. 2). An additional insertion (Gly503-Gly528) of PsaA was found to be disordered at the lumenal side in the far-red PSI. In the PsaL subunit, loop3 (Thr8-Leu28) and loop4 (Gln117-Pro134) are inserted in the far-red PSI (Supplementary Fig. 11), which are located at the stromal side in the structure

(Fig. 2). Interestingly, all these insertions and deletion (loop1-4) are located at one side (PsaA-PsaL side) of the structure of a PSI monomer (Fig. 2a). Both loop1 and loop4 do not interact directly with other subunits (Fig. 2b, d), whereas loop3 is elongated to the interface between PsaA and PsaD, and contributes to the structural stability of loop2 (Fig. 2c). Chl a865 (a865A) is bound to a region in white PSI that is occupied by loop2 in far-red PSI, and it is lost in the far-red PSI. Apparently, the insertion of loop2 hindered the binding of Chl a865A (some of the numbering of pigments and other cofactors discussed in the text are different from those registered in the PDB file; for a complete correspondence of the numbering of pigments and cofactors in the text with those in the PDB file, refer to Supplementary Table 4) in the far-red PSI (Fig. 2c). The loop2 is conserved in the *psaA* gene expressed under far-red light condition but absent in the gene expressed under white light condition in the Chl *f*-containing species (Supplementary Fig. 8). This loop2 is also absent in PsaA from *Synechocystis* sp. PCC6803 (NCBI: WP_010872067) and *T. elongatus* (NCBI:WP_011056578.1), from which the PSI structure has been solved and the corresponding Chl a865A molecule is present. These results indicate that the far-red light-induced expression of different copies of the PSI genes *psaA, psaB, psaI, and psaL*, and differences in the sequences of these genes, especially the *psaA* gene, caused the structural changes of the PSI complex, especially with respect

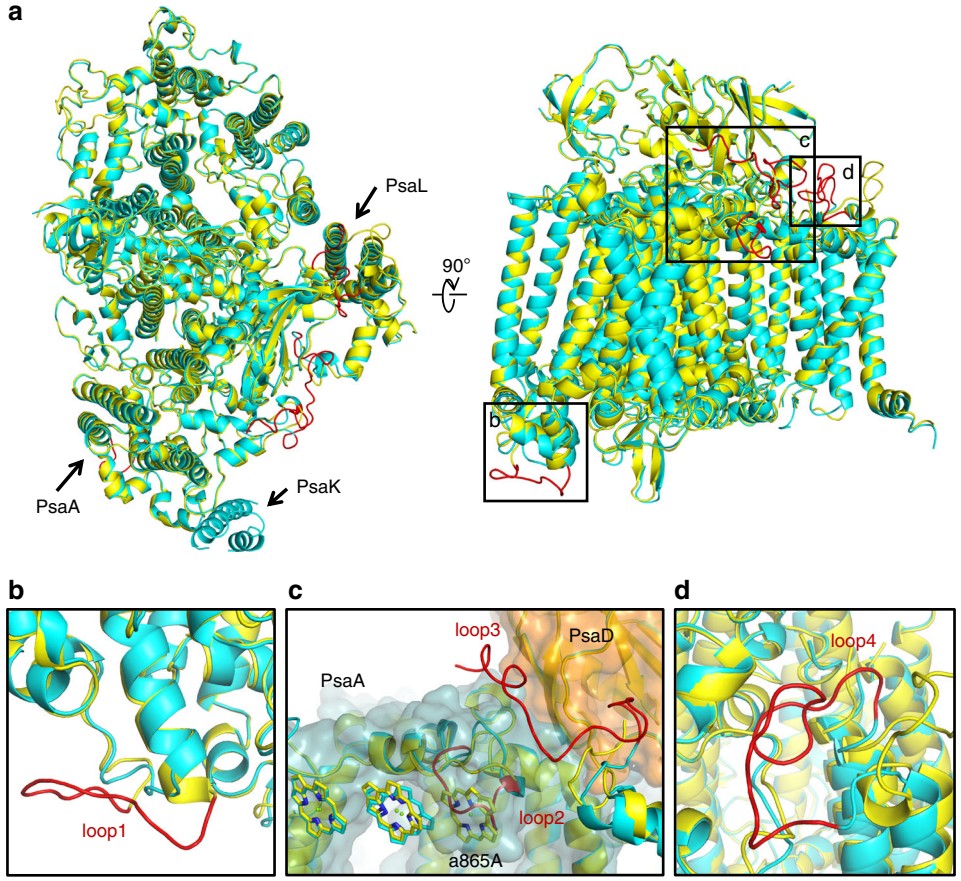

**Fig. 2 Superposition of the structures of the white PSI and far-red PSI monomers. a** Superposition of the white PSI (yellow) and far-red PSI (cyan) viewed along the membrane normal from the stromal side. The structural differences (loop insertions) are colored in red. The RMSD is 0.62 Å for 1855 Cα atoms from subunits commonly observed between white PSI and far-red PSI. **b–d** Close-up view of the loop insertions at A232-D244 (loop1) of PsaA in white PSI **b**, P331-N338 (loop2) of PsaA in far-red PSI **c**, T8-L28 (loop3) of PsaL in far-red PSI **c**, and Q117-P134 (loop4) of PsaL in far-red PSI **d**. **c** PsaA (dark cyan) and PsaD (orange) are also shown as surface model.

to the insertion (and deletion) of the four loop regions in the far-red PSI (Fig. 2 and Supplementary Figs. 8–11).

The cofactors of the white PSI monomer contained 90 Chl $a$, 16 β-carotene (BCR), 3 [4Fe-4S] cluster, 2 phylloquinones, and three lipid molecules (Supplementary Table 5). The locations of the cofactors are similar to those of the Chl $a$-containing cyanobacterial PSI[6,11]. However, we identified seven molecules of Chl $f$ (f826A, f827A, f830A, f832A, f844A, f810B, and f825B) in the monomer of the far-red PSI based on the difference density maps calculated between the cryo-EM experimental maps of the white PSI and that of the far-red PSI, as well as between the cryo-EM experimental map of the far-red PSI and the calculated density map derived from the atomic models of the far-red PSI with all Chls assigned as Chl $a$ (see Methods for details) (Fig. 3a and Supplementary Fig. 12). Other cofactors in the far-red PSI include 83 Chl $a$, 16 β-carotene, 3 [4Fe-4S] cluster, 2 phylloquinones and 2 lipid molecules in a monomer. This gives rise to a Chl $a/f$ ratio of 11.8 in the far-red PSI, which is very similar to the results reported recently[31]. The relative areas after taking into consideration the molecular coefficients of Chl $a$ and Chl $f$[32] in the 80 K absorption spectrum of far-red PSI were 89.1 and 10.9, respectively (Supplementary Fig. 13a, insert). This value is also consistent with the number of Chls $f$ found in the structure of the far-red PSI. Compared with the white PSI structure, the far-red PSI has one additional Chl $a$ (a101K) in the PsaK subunit, as PsaK was lost in the white PSI. However, one Chl $a$ (a865A) is depleted in the far-red PSI as described above. As a result, the

total number of Chls are the same between the white PSI and far-red PSI.

As no novel binding site of Chl $f$ was found, Chl $f$ has substituted the binding sites of Chl $a$ upon far-red light induction in the far-red PSI. In agreement with the previous studies, the RC Chls were a pair of Chl $a$/Chl $a'$ (epimer of C13²–Chl $a$) in both white and far-red PSI[6]. However, Chl a865A was found in the white PSI but absent in the far-red PSI. As this Chl is located at the interface of monomers in the white PSI trimer, its absence in the far-red PSI trimer may suggest a different energy transfer behavior among monomers between the far-red PSI and white PSI trimers.

**Binding sites of Chl $f$ and their possible functions.** The seven molecules of Chl $f$ identified in the far-red PSI are located at the interface of each PSI protomer (Figs. 3a and 4a). Among them, five Chl $f$ (f826A, f827A, f830A, f832A, and f844A) are coordinated by PsaA and the other two Chl $f$ (f810B and f825B) are coordinated by PsaB. Except for one Chl $f$ (f825B), five (f826A/ f827A/f830A/f832A/f844A) form a Chl $f$ network in a monomeric unit with edge-to-edge distances of 10.0–13.3 Å between the chlorin rings (Fig. 4). In addition, Chl f810B is also located at the same PsaA side with an edge-to-edge distance of 18.0 Å to Chl f844A. These two Chl $f$ molecules are coupled through an intermediate BCR molecule BCR102I (Fig. 4c) (see below). Thus, Chl f810B can be considered to be part of the Chl $f$ network.

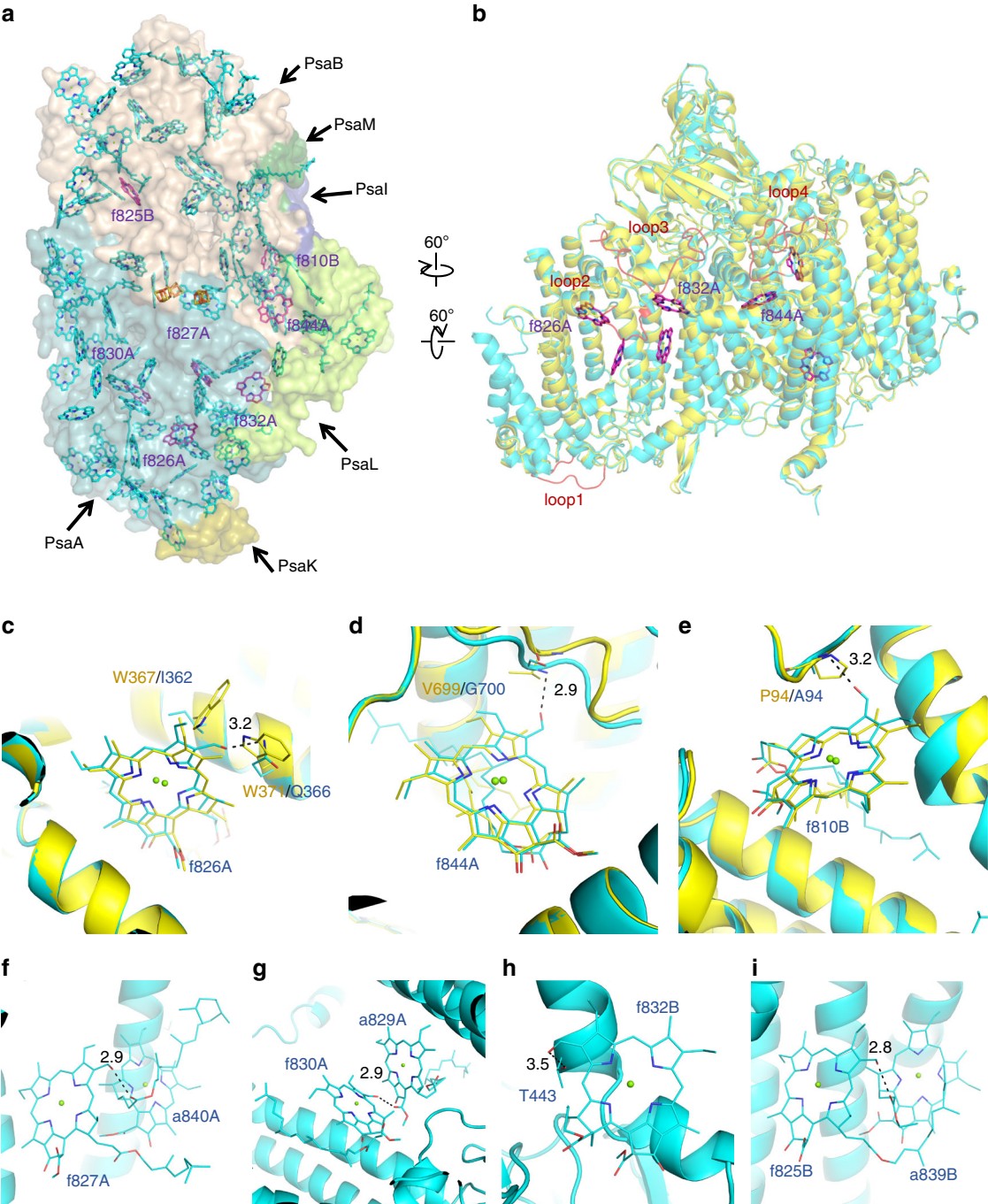

**Fig. 3 Arrangement of pigments in the far-red PSI monomer. a** Arrangement of the pigments in the far-red PSI monomer with protein subunits depicted in surface models, with a viewed along the membrane normal from the stromal side. For clarity, PsaC, PsaD, and PsaE are not shown. Chls *f* are shown as magenta stick (the color of the Chl *f* molecules f826A/f827A/f830A/f832A look like purple owing to the overlap with the cyan color of the surface model of PsaA) and the other ligands are shown as cyan stick. **b** Arrangement of the Chls *f* in the far-red PSI monomer with the proteins depicted in cartoon models. The protein structures of white PSI (yellow) and the far-red PSI (cyan) are superposed with each other. The insertion loops found in far-red PSI are colored in red. Chls *f* are shown as magenta stick. **c–e** Close-up view of the binding environments for f826A **c**, f844A **d** and f810B **e**. The structure of white PSI (yellow) and the far-red PSI (cyan) are superposed with each other and shown as cartoons. Chls and residues different between white PSI and far-red PSI are shown as sticks. **f–i**, Close-up view of the binding environments for f827A **f**, f830A **g**, f832A **h**, and f825B **i**. The structure of the far-red PSI (cyan) is shown as cartoon, and Chls are shown as sticks.

Among the seven Chl *f* molecules, three (f826A, f832A, and f844A) are located near loop2–4 (Fig. 3b), suggesting the close correlation between the substitution of Chl *a* by Chl *f* and the structural changes occurred in the loop regions in the far-red PSI. At the Chl f826A-binding site, W367 and W371 of PsaA in the white PSI are changed to I362 and Q366 with relatively smaller side chains in the far-red PSI (Supplementary Fig. 8); among them, Q366 is hydrogen-bonded to the oxygen atom of the formyl group of f826A with a distance of 3.2 Å (Fig. 3c). At the f844A-binding site, V699 of PsaB in the white PSI is changed to G700 in the far-red PSI (Supplementary Fig. 9), with its main chain nitrogen associated with the oxygen atom of the formyl

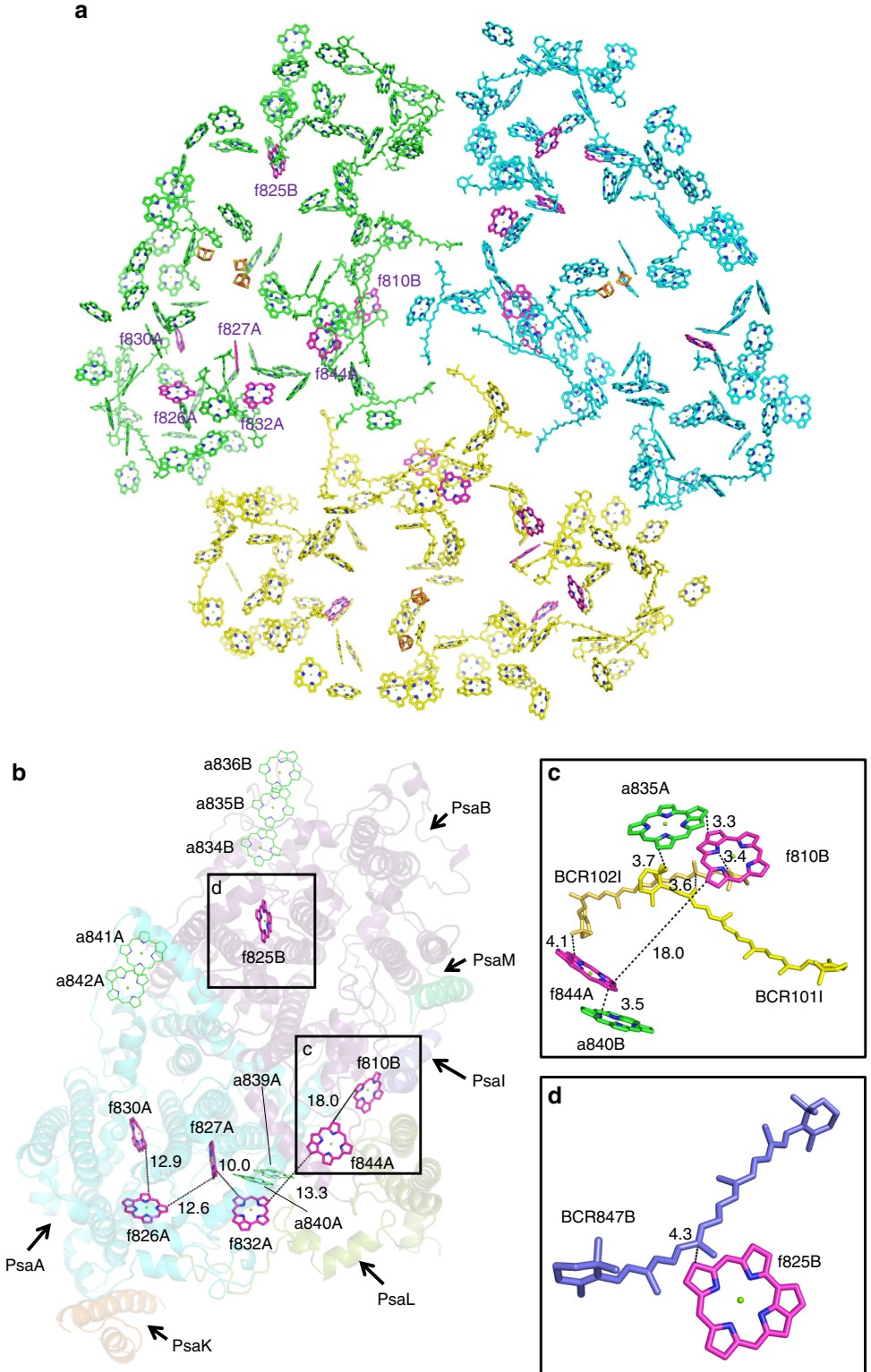

**Fig. 4 Arrangement of pigments in the far-red PSI trimer. a** Arrangement of the pigments in a far-red PSI trimer viewed along the membrane normal from the stromal side. The Chls *f* are colored in magenta, and other cofactors are colored in cyan, yellow and green in three monomers, respectively.
**b** Arrangement of Chl *f* in a far-red PSI monomer viewed along the membrane normal from the stromal side, with the protein subunits depicted in a cartoon model. For clarity, PsaC, PsaD, and PsaE are not shown in the model. Chls *f* and several Chls *a* that are assumed to form red Chls are colored in magenta and green, respectively. **c**, **d** Close-up views of nearby pigments of f844A and f810B **c**, f825B **d**.

group of f844A at 2.9 Å (Fig. 3d). At the f810B-binding site, P94 of PsaB in the white PSI is changed to A94 in the far-red PSI (Supplementary Fig. 9), and its main chain nitrogen is associated with the formyl group at 3.2 Å (Fig. 3e). All these changes result in residues with rather small side chains in the far-red PSI, enabling them to be able to accommodate Chl $f$ which has an formyl group in its chlorin ring instead of a methyl group in Chl $a$. These residues related to the Chl $f$-binding in PsaA and PsaB, are conserved in the two genes expressed under far-red light, but not in the white light, in the Chl $f$-containing species (Supplementary Figs. 8 and 9). These results indicate that these changes in the amino-acid sequences are important to accommodate the formyl group of the three Chl $f$ molecules appeared under far-red light conditions. On the other hand, no changes in the amino-acid residues are observed in the binding sites for f827A, f830A, f832A, and f825B, and the formyl groups of these Chl $f$ are associated with an oxygen atom of Chl a840A phytol (2.9 Å), an oxygen atom of the C13$^2$ group of Chl a829A (2.9 Å), a hydroxyl group of Thr443 of PsaA (3.5 Å), and an oxygen atom of Chl a839B phytol (2.8 Å), respectively (Fig. 3f–i).

It is well-known that cyanobacterial PSI core binds red-shifted Chls (red Chls) whose energy levels are lower than P700. It has been suggested that red Chls consist of 7–11 molecules of Chl $a$ per 96 Chl $a$ based on the absorption spectra of PSI at cryogenic temperatures[33–35]. At least four Chl $a$ clusters are necessary to explain the absorption spectrum of red Chls. These red Chl $a$ components are also seen in white and far-red PSI in the 80 K absorption spectra (Supplementary Fig. 13). Several red Chl $a$ candidates have been reported based on hole-burning analysis[33]. In the present structure, Chls a836B–a835B–a834B and a841A–a842A form two Chl $a$ clusters, and thus can be considered as candidates for the red Chl $a$ clusters (Fig. 4b). These Chls $a$ are conserved between white and far-red PSI. The triplet Chls a836B–a835B–a834B are also found in the *T. elongatus* PSI structure, and they are adjacent to PsaX but not directly related to PsaX[6]. This "red trap" may therefore also work in the *H. hongdechloris* PSI. Another candidate for the red Chl $a$ is a839A–a840A[33] (Fig. 4b), which is also conserved between white and far-red PSI. These two Chls $a$ (a839A–a840A) are very close to f832A and f827A, suggesting that this red Chl $a$ cluster is closely associated with the six Chl $f$ network and these Chls may share the excitation energy. The other red Chl $a$ candidates reported are Chls a840B–a844A and a810B–a835A. However, a844A and a810B are changed to f844A and f810B in far-red PSI, respectively. As a result, a840B and a835A also take part in the six Chl $f$ network. This allows a835A and f844A to interact through BCR102I (Fig. 4c). Among them, f810B interacts with a835A at a distance of 3.3 Å, and Chl a840B interacts with f844A at a distance of 3.5 Å with an oblique dipole orientation of the $Y$ axis between the two Chl molecules (Fig. 4c). The close Chl–Chl interaction and their dipole orientation appear to lower their energy levels, suggesting that the two Chl $a/f$ hetero-dimers are the fluorescence-emitting Chls at 813 nm (Supplementary Fig. 13). One Chl $f$, f825B, is far from the Chl $f$ network (Fig. 4b), and interacts with BCR847B at a distance of 4.3 Å (Fig. 4d). This seems to be a Chl-Car quencher by either charge transfer or excitation energy transfer[36]. These structural and spectroscopic observations suggest that Chl $f$ molecules are responsible for excitation energy transfer and quenching events under far-red light conditions.

The absorption spectrum of the far-red PSI shows at least three components at long wavelength region, which are located at 737, 752, and 795 nm (Supplementary Fig. 13). Our time-resolved fluorescence study of the far-red PSI clearly showed that Chl $f$ molecules serve as excitation energy transfer components in the early time range of 5–300 ps after excitation. In particular,

fluorescence components appeared at 746, 752, 753, and 813 nm at 77 K can be ascribed to Chl $f$ (Supplementary Fig. 13). This indicates that the red Chls $a$ and Chls $f$ are equilibrated within this time range. Thus, Chl $f$ molecules function in uphill energy transfer for energy-trapping at the RC Chls. The candidates for the fluorescence components at 746–755 nm are likely f826A/ f827A/f830A/f832A/f844A and f825B, suggesting the contribution of these Chl $f$ to the uphill energy transfer to the RC Chls at physiological temperatures[20,27,28].

It has been proposed that Chl at the position of A$_{-1}$ site is Chl $f$ based on spectroscopic measurements[16]. However, we found that all of the seven Chl $f$ molecules are located at the periphery region of the PSI core as well as in the interfacing regions among the PSI monomers, and all RC Chls, including the special pair, A$_{-1}$, and A$_0$, are Chl $a$ molecules in the far-red PSI. These observations indicate that Chl $f$ molecules may not be involved in the charge transfer and electron transfer reactions directly, but function in either uphill energy transfer and/or quenching.

## Discussion

Some cyanobacteria have been found to undergo extensive remodeling regarding the expression of some photosynthetic genes upon transfer from white light to far-red light, which is accompanied by the synthesis of Chl $f$ and its incorporation and functioning in the photosystems[3–5,21–24,29,31]. The present structural analyses indeed revealed the changes of the sequences and structures of four subunits, PsaA, PsaB, PsaI, and PsaL, between white PSI and far-red PSI. In addition, the genes for PsaF and PsaJ were reported to be also different between cyanobacteria grown under white and far-red light conditions. These subunits were not resolved in the cryo-EM structure of both white and far-red PSI, owing to either weak densities (white PSI) or almost absence of their densities (far-red PSI). As these subunits are apparently present in the solution samples of both PSI, they may be lost during the grid preparation. This suggests their loose association with the PSI core, which may facilitate the structural remodeling of PSI upon transition between the white and far-red light conditions. Thus, the loose association of these subunits with the PSI core may be a structural feature for cyanobacteria that undergo remodeling under different light conditions and use Chl $f$ upon transfer to far-red light conditions.

Both white and far-red PSI contained 90 Chls, which is comparable to the 96 Chls identified in the crystal structure of PSI from *T. elongatus*[6] if we consider the loss of several subunits in the structure of both white and far-red PSI. Among these Chls, we identified seven Chl $f$ in the far-red PSI based on the approach of difference density maps between the cryo-EM maps of white and far-red PSI as well as between the experimental cryo-EM map of the far-red PSI and that calculated from the PSI structure with all Chls assigned as Chl $a$. This number agrees well with the ratio of 1:8 for Chl $a/f$ found in this cyanobacterium grown under far-red light conditions[31], and is also comparable with the spectral analysis performed in this study. As all of the Chl-binding sites are conserved between white and far-red PSI (except Chl a865A), seven Chl $a$ are replaced by Chl $f$ in the far-red PSI. Sequence and structural analysis showed that among the seven binding sites for Chl $f$, at least three are made possible by remodeling of the sequences and hence the structure of PSI subunits upon transition from white to far-red light conditions. These three Chl $f$ are f826A, f844A, and f810B, and two of them are located near loops 2–4 (Fig. 3b) in the PsaA and PsaL subunits that have distinctly different sequences and structures between the genes expressed in the white and far-red light conditions. Importantly, the residues coordinating these three Chl $f$ are all changed from large ones in the white PSI to smaller ones in the far-red PSI, and these smaller

residues are conserved among the genes expressed under far-red light but not in the genes expressed under white light conditions. This illustrates the need to accommodate Chl $f$ in these binding sites, as the methyl group at the $C_2$ position on the chlorin ring of Chl $a$ is replaced by a slightly larger (and hydrophilic) formyl group in Chl $f$. No apparent differences were found for the binding sites of the remaining four Chl $f$ molecules between the white PSI and far-red PSI, suggesting the flexibility of these sites in accommodating either Chl $a$ or Chl $f$.

Among the seven Chl $f$ found in the far-red PSI, six (except Chl f825B) form a cluster with rather short inter-molecular distances (Chl f810B is included in this cluster because of its interaction with Chl f844A through an intermediate BCR molecule). This cluster is also close to a Chl $a$ dimer Chl a839A/a840A, which has been suggested to serve as a red Chl pair. Thus, these six Chl $f$ may serve as energy trap for transfer to the RC Chls or quenching of excess energy under high light conditions. These Chls $f$ are located in the periphery region of the PSI core and not related with the ETC. The remaining one Chl $f$ (Chl f825B) is also located at a site not related with the ETC. These results suggest that Chl $f$ induced by the far-red light is not involved in the PSI electron transfer directly. However, time-resolved spectroscopic measurements have suggested that one or both Chl $A_{-1}$ may be Chl $f$ in the far-red PSI[16,18]. In the light-induced absorption changes of P700 measured with $T.\ elongatus$ PSI and far-red PSI (Supplementary Fig. 7), we did not observe signals reflecting the electrochromic shift of Chl $f$. As the six Chls involved in PSI ETC, namely, P700, $Chl_{A-1}$, and $Ch_{A0}$ located in PsaA and PsaB, respectively, form an excitonically coupled state[37], our result is consistent with the structure that Chl $f$ is not involved in PSI ETC directly.

In summary, we determined the structure of PSI trimers from $H.\ hongdechloris$ grown under either white or far-red light conditions by cryo-EM at high resolutions, and identified seven Chl $f$ in a PSI monomer from cells grown under the far-red light condition. These Chls $f$ are associated at the periphery of the PSI core, and five of them are coordinated by PsaA, whereas two are coordinated by PsaB. In correlation with the appearance of Chl $f$, at least four PSI subunits (PsaA, PsaB, PsaI, PsaL) were expressed from different genes between white and far-red light conditions. The structures of these four subunits are also slightly different between white PSI and far-red PSI. In particular, three loop regions distributed over PsaA and PsaL, together with some specific residues in PsaA and PsaB, were found to be important for the binding of three Chl $f$ molecules, whereas the binding sites for the other four Chl $f$ molecules are not much different between white PSI and far-red PSI. All seven Chl $f$ are located at the periphery region of the PSI core but not related with the ETC directly. These results indicate that Chl $f$ functions to enhance the uphill energy transfer and/or energy quenching, instead of the initial charge transfer.

## Methods

**Purification of the PSI from $H.\ hongdechloris$.** We cultured the cells of $H.\ hongdechloris$ under white (HITACHI FL20 SSD) and far-red (Edison OPTO Corporation, Cherry Red) lights[20] in a 7-liter artificial seawater-IMK medium containing a final concentration of 25 mM Tris (pH 8.0), 11.75 mM NaNO₃, 0.15 mM $K_2HPO_4$ at 303 K with stirring. The cells were harvested and disrupted with glass beads (diameter 0.1 mm) on ice in the dark for 16 cycles, with 10 s on, and 215 s off for each cycle. The unbroken cells were removed by twice centrifugation at $3700 \times g$ for 7 min at 275 K. The supernatant was collected and centrifuged at $39,900 \times g$ for 20 min at 275 K to pellet the membranes. The precipitate was suspended in a buffer containing 50 mM MES-NaOH (pH 6.0), 25% (w/v) glycerol, 10 mM MgCl₂, and 5 mM CaCl₂ (buffer B) as thylakoid membranes. The thylakoid membranes were solubilized by 2% dodecyl-β-D-maltoside (β-DDM), followed by precipitation and resolubilization of the pellet by 2% β-DDM. Then, the supernatant was loaded onto an anion exchange column (UnoQ, BioRad) equilibrated with 10% (w/v) glycerol/ 50 mM MES (pH 6.0)/5 mM CaCl₂/

30 mM NaCl/0.04% DM. The PSI complexes were eluted with a 30–400 mM NaCl linear gradient. Fractions containing PSI complexes were collected and subjected to sucrose density gradient (5–20% in 50 mM MES(pH 6.0), 5 mM CaCl₂, 0.04% (white PS1) or 0.1% (far-rede PS1) β-DDM) centrifugation twice to remove contaminating PSII and a small amount of monomeric PSI, and the final samples were collected and concentrated by a membrane filter Ultra-amicon (MWCO = 100 kDa). These procedures were summarized in Supplementary figure 1. For analysis of protein composition, PSI complexes were solubilized with 5% sodium dodecyl sulfate and 2% 2-mercaproethanol for 30 min at room temperature, followed by separation with a 16–22% acryl amide gradient gel containing 7.5 M urea[38]. Gels were stained with Coomassie Brilliant Blue (CBB) R-250.

**Assignments of polypeptides by mass spectrometry.** After SDS-PAGE, each band was excised and de-stained, followed by reduction and alkylation by dithiothreitol and acryl amide. The gel slices were digested with trypsin (TPCK treated; Worthington Biochemical Co.) at 37 °C for 12 h. The resultant peptides were analyzed using a Q Exactive mass spectrometer (Thermo Fisher Scientific GmbH, Bremen, Germany) coupled with an Easy nLC 1000 (Thermo Fisher Scientific GmbH, Bremen, Germany). Separation of peptides was carried out with a linear gradient from 0 to 80% solvent B (100% acetonitrile /0.1% formic acid) over 20 min at a flow rate of 300 nL min⁻¹ using a nano-ESI spray column (100-mm length × 75-μm internal diameter, 3-μm opening, NTCC analytical column C18, Nikkyo Technos). The spectra were obtained using a data-dependent top 10 method and were searched against the in-house database including $H.\ hongdechloris$ sequences with Mascot Version 2.6 software (Matrix Science) using the following parameters: Type of search, MS/MS ion search; Enzyme, Trypsin; Fixed modification, none; variable modifications, acetyl (Protein N-term), formyl (Protein N-term), Gln- > pyro-Glu (N-term Q), oxidation (M), propionamide (C), Deamidated (NQ); Mass values, monoisotopic; Peptide Mass Tolerance: ±15 ppm; Fragment Mass Tolerance: ±20 mmu; Max Missed Cleavages: 3; Instrument type, ESI-TRAP[39].

**Assignment of polypeptides by N-terminal sequencing.** After SDS-PAGE, polypeptides were blotted onto a polyvinylidene difluoride membrane and stained with CBB. The CBB-stained bands were cut out and subjected to the Edman degradation using a Procise HT protein sequencing system (Applied Biosystems, Foster City, CA, USA). The N-terminal sequence of each band was read up to 15 amino-acid residues[40].

**Spectroscopic measurements.** Absorption spectra were recorded using a spectrophotometer (JASCO V-660, Japan) at 80 K with a cryostat for liquid nitrogen temperature (OptistatDN, Oxford Institute, Oxford, UK) and an Oxford ITC-601PT controller. Steady-state fluorescence spectra were recorded using a spectrofluorometer (JASCO FP-8500, Japan) with a commercial Dewar system at 77 K. The spectral sensitivity of the fluorometer was corrected using the radiation profile of the standard lamp. To measure the low-temperature fluorescence spectrum, polyethylene glycol (average molecular weight 3350, final concentration 15% (w/v), Sigma-Aldrich) was added to obtain a homogeneous glass. Time-resolved fluorescence spectrum and fluorescence decay curves were measured using time-correlated single-photon counting methods, with an excitation wavelength of 425 nm (i.e., the second harmonic of the 850-nm oscillation). Fluorescence lifetime was estimated via the convolution calculation[20]. All pigments were simultaneously excited using an excitation wavelength of 425 nm. For measurement of the light-minus-dark spectra of P700, samples were suspended in 0.04% DM, 1 mM ascorbate, 200 μM methyl viologen, 25 μM $N',N',N',N'$-tetramethyl-$p$-phenylenediamine and 50 mM Tris-HCl (pH 8.0), and illuminated with white light[40].

**Cryo-EM data collection.** For cryo-EM analysis, the samples of the white PSI and far-red PSI were adjusted to similar conditions, e.g., 32 μg of Chl mL⁻¹ (white PSI) and 71 μg of Chl mL⁻¹ (far-red PSI) in a buffer containing 50 mM MES (pH 6.0), 5 mM CaCl₂, 0.2–2.0% sucrose, and 0.01% β-DDM. Aliquots of 3 μL of the white PSI and far-red PSI samples were applied to glow-discharged holey carbon grids (Quantifoil R2/1, Mo 300 mesh) covered with 5–10 nm amorphous carbon film. The grids were incubated for 30 s and then washed once with 2 μL wash buffer (50 mM MES (pH 6.0), 5 mM CaCl₂) without sucrose and β-DDM using Vitrobot Mark IV (Thermo Fischer Scientific) at 4 °C and 100% humidity. This wash process to remove sucrose and β-DDM dramatically enhanced the image contrast of the particles[15]. After blotting with filter papers for 3 s, the grids were immediately plunged into liquid ethane cooled by liquid nitrogen and then transferred into a cryo-electron microscope (Titan Krios, Thermo Fischer Scientific) equipped with a field emission gun, a Cs corrector (CEOS GmbH), and a direct electron detection camera (Falcon 3EC, Thermo Fischer Scientific). The microscope was operated at 300 kV and a nominal magnification of ×75,000 (pixel size of 0.87 Å). Movies were recorded using the Falcon 3EC camera in a linear mode. The nominal defocus range was −1.25 to −3.0 μm. Each exposure of 2.5 s (total electron dose 47 electrons Å⁻²) was dose-fractionated into 33 movie frames. A total of 4754 and 5712 micrographs were obtained for the white PSI and far-red PSI trimers, respectively.

**Cryo-EM image processing**. The movie frames obtained were aligned and summed using the MotionCor2 software[41] to obtain a final dose-weighted image. Estimation of the contrast transfer function (CTF) was performed using the CTFFIND4 program[42]. All of the following processes were performed using RELION[43]. For structural analyses of the white PSI trimer structure, 2,288,744 particles were automatically picked from 4754 micrographs and then used for reference-free 2D classification. Then, 1,224,569 particles were selected from the good 2D classes and subjected to 3D classification without imposing any symmetry. The initial model for the first 3D classification was generated de novo from 2D classification. As shown in the Supplementary Fig. 2c, the white PSI trimer structure was reconstructed from 546,366 particles at an overall resolution of 2.35 Å. For structural analyses of the far-red PSI trimer, 2,288,744 particles were automatically picked from 5712 micrographs and used for reference-free 2D classification. Then, 1,425,999 particles were selected from the good 2D classes and subjected to 3D classification without imposing any symmetry. The initial model for the first 3D classification was generated de novo from 2D classification. As shown in the Supplementary Fig. 3c, the far-red PSI trimer structure was reconstructed from 311,993 particles at an overall resolution of 2.41 Å. All of the resolution was estimated by the gold-standard FSC curve with a cutoff value of 0.143 (Supplementary Fig. 5). The local resolution was estimated using RELION[43].

**Model building and refinement**. The 2.35-Å and 2.41-Å Cryo-EM maps were used for model building of the white PSI and the far-red PSI trimers, respectively. First, the crystal structure of *T. elongatus* PSI (TePSI, PDB codes: 1JB0) was manually fitted into the 2.35-Å and 2.41-Å cryo-EM maps using UCSF Chimera[44], and then the structures were inspected and adjusted individually with COOT[45]. The amino-acid sequences of the *T. elongatus* PSI structural models were then mutated to its counterparts from *H. hongdechloris* under white and far-red light conditions, respectively. The complete structures of the white PSI and the far-red PSI trimers were then refined with phenix.real_space_refine[46] with geometric restraints for the protein–cofactor coordination. The final models were further validated with MolProbity[47] and EMringer[48]. The statistics for all data collection and structure refinement are summarized in Supplementary Table 1.

**Difference map analysis of the far-red PSI trimer**. To identify the positions of Chl *f* in the far-red PSI trimer, two difference maps were calculated (Supplementary Fig. 12). First, a difference map was calculated by subtracting the map calculated based on the refined atomic model without Chl *f* but with all Chls assigned as Chl *a* from the experimental cryo-EM map of far-red PSI, i.e., (EM map) minus (model map). The map was calculated using PHENIX[46] with the command line option "phenix.real_space_diff_map" available in the 2017-07-07 nightly build version. In order to exclude pseud-positive signals in the first difference map, a second difference map was calculated by subtracting the cryo-EM densities in the white PSI from the densities in the far-red PSI in the areas where Chl *f* molecules are found, i.e. (far-red PSI cryo-EM map) minus (white PSI cryo-EM map). Each of the rotational and translational matrix was calculated based on the refined atomic coordinates using lsqkab in CCP4[49]. Each Chl *a* map in the white PSI was superposed with the corresponding Chl map in the far-red PSI with calculated rotational and translational matrix using maprot in CCP4[49]. Chl *a* maps in the white PSI and the corresponding Chl maps in the far-red PSI were normalized based on the ratio of the root mean square map density value, and then the difference maps for all Chl were calculated using UCSF Chimera[41] with the command line option "vop subtract". The Chls *f* molecules were assigned based on the residual densities of both difference maps with a threshold of more than 5σ (Supplementary Fig. 12). The difference map analysis shows two groups of Chl *f* (one is f826A/f844A/f810B and the other one f827A/f830A/f832A/f825B) based on the noise level of the residual densities. At the first group of the Chls *f*-binding sites, some changes of the surrounding amino acids were found which are correlated with the binding of Chl *f*. This resulted in a lower noise in the densities because of a high occupancy of Chl *f*. In contrast, at the second group of the Chls *f*-binding sites, no changes were found in the amino-acid residues around the Chl *f*-binding sites, which resulted in a higher noise level in the density because of a lower occupancy of Chl *f*.

**Reporting summary**. Further information on research design is available in the Nature Research Reporting Summary linked to this article.

## Data availability
Atomic coordinates and cryo-EM maps for the reported structure of white PSI timer and far-red PSI trimer have been deposited in the Protein Data Bank under accession codes 6KMW [https://www.rcsb.org/structure/6KMW] and 6KMX [https://www.rcsb.org/structure/6KMX], respectively, and in the Electron Microscopy Data Bank under accession codes EMD-0726 and EMD-0727, respectively. Other data are available from the corresponding authors upon reasonable request.

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

## Acknowledgements

This work was supported by the Platform Project for Supporting Drug Discovery and Life Science Research (Basis for Supporting Innovative Drug Discovery and Life Science Research (BINDS)) from AMED (to N.M.), PRESTO from JST Grant No JPMJPR16P1 (to F.A.), JSPS KAKENHI No. JP17K07442, JP19H04726 (to R.N.), JP17H06434 (to J.-R.S.), No. 17726220801, 17K07453, 18H05177 (to T.T.), JP19K22396 (to F.A.). S.I.A., S.A., T.T. and J.-R.S. gratefully acknowledge the JSPS-RFBR grant (No. 19-54-50002). T.T. would like to thank Prof. T. Noguchi for valuable discussion.

## Author contributions

K.K., R.N., N.M., S.I.A, F.A., J.-R.S. and T.T. conceived and coordinated the project. T. Shinoda purified the PSI and performed their biochemical and spectroscopic characterizations. S.A. performed spectroscopic analysis. T. Suzuki., N.D. and M.C. identified the protein subunit by gene and MS analyses. N.M., F.A. and R.N. collected cryo-EM images. K.K. and N.M. processed the EM data. K.K. reconstructed the final EM maps. K.K. built the structure model. K.K. refined the final models. K.K and N.M. analyzed the structure. T.Shinoda, R.N., S.A., S.I.A and T.T. proposed structural interpretation on the basis of spectroscopic analyses. K.K., T.Shinoda, R.N., F.A., N.M., J.-R.S. and T.T. wrote the paper. All authors contributed to the interpretations of the results and improvement of the manuscript.

## Competing interests

The authors declare no competing interests.
