## [Peer Review File · Nature Communications]

Reviewers' comments:

Reviewer #1 (Remarks to the Author):

This manuscript addresses important questions of the association of chlorophyll f that absorbs light at the far-red spectrum in PSI that usually utilizes chlorophyll a for light absorption and conversion into chemical energy. The experimental systems were chosen very well and studied by advanced biochemical and structural methods. The main tool was cryo-EM technique the interpretation of which is frequently abused. This is certainly not the case of this manuscript that is well executed, interpreted and written.

The model building described in the manuscript is highly satisfactory. The authors correctly refrained from including PsaJ, PsaF and PsaK in the model. "The poor densities of PsaF and PsaJ may be due to the lack of the PsaX subunit which is not found in the genome of *H. hongdechloris*, because PsaF and PsaJ are located near PsaX in the PSI-trimer structure and therefore their binding may become weak." However the explanation is questionable because *Synechocystis* PSI contains no PsaX12. Moreover a novel chlorophyll molecule that is present in the same space complementing a triplet that is a good candidate for a "red trap". It is advisable to mention red traps and their distribution in the periphery of PSI, in relation to up-hill excitation transfer from chlorophyll f. They clearly see the [2-formyl]-chlorophyll a and certainly it would not escape detection if present in the ETC where the local resolution is exceptionally high. The chlorophyll f molecules are apparent in the periphery and clearly detected at map densities of higher than 5 sigma. This assessment should be taken as a gold standard for all other cryo-EM structures. The sentence "we found no evidence for participation of chlorophyll f in the ETC" should be included in the abstract not only in the last sentence of the manuscript.

I would highly recommend accepting this manuscript for publication.

Reviewer #2 (Remarks to the Author):

The manuscript by Kato et al. describes two cryo-EM structures of PSI core from *H. hongdechloris* grown under white and far-red light conditions. The two structures are solved at overall resolutions of 2.35 and 2.41 Å, respectively. The white PSI contains only Chl a, while the far-red PSI contains Chl f in addition to Chl a, which was induced by the far-red light. Furthermore, the authors observed the replacement of four core subunits with the homologous gene products in the far-red PSI, and identified the role of specific protein subunits and fragments in Chl f binding. The cryo-EM data are sound, and the high resolution map allows the accurate model building. The work would be a great addition to the photosynthesis field, however, I think the study also has some weaknesses as listed below.

The first concern is that the authors did not give enough information in the Method section. For example, the description of the PSI purification is very short and does not give any details. The experimental conditions for the measurement of absorption and fluorescence spectra were not described at all. The method section should be described in more details. In addition, I suggest the authors to provide an additional ED figure to show the characterization of both PSI samples, including the SDS-PAGE, the profiles of chromatography and sucrose density gradient centrifugation.

It is a big shortcoming that the PSI core is incomplete in both structures. As the authors described in the text, the white PSI core is loosely associated with PsaK, PsaF and PsaJ, which were therefore deleted in the final model. The authors explained the possible reason for the poor densities of PsaF and PsaJ, and attributed it to the lack of the PsaX subunit. However, the explanation is not convinced to this reviewer. The PSI from *Synechocystis* 6803 does not contain PsaX either, nevertheless, the subunits PsaF and PsaJ are well behaved in the crystal structure (Malavath, BBA, 2018). I believe that the partial loss of the three subunits, especially PsaK (this subunit is present in the far-red PSI), in the white PSI sample is due to the high concentration of β -DDM (0.1%) used for the grid preparation. I'm not sure if the sample was further concentrated before this process (no description in the Method). If so, the β -DDM concentration could be even higher. It is quite

common for membrane protein complexes to dissociate and lose some peripheral subunits under conditions of high concentration of detergent. The authors need to explain why they used the higher concentration of β -DDM in grid preparation for the white PSI than that for far-red PSI (0.04%).

The far-red PSI contains PsaK, but does not bind PsaF and PsaJ at all. It is not clear if these subunits were lost during purification or grid preparation processes, or they are truly absent in PSI from the beginning. If the subunits PsaF and PsaJ are indeed absent from far-red PSI, the physiological relevance of the loss of PsaF/PsaJ and the adaptation to the far-red light should be discussed. PsaF was previously suggested to be involved in binding electron donor, such as the c-type cytochrome. If the far-red PSI does not contain PsaF, the P700 reduction activity might be affected. The measurement of activity of both white and far-red PSI should be done.

The authors assigned seven Chl f molecules in each PSI monomer. Based on the cryo-EM densities shown in Extended Data Figure 6, the identity of Chls f826A, f844A, f810B is convincing. However, I have some doubts about the assignment of other four Chl f molecules. Since the difference densities for the 2-formyl groups of these four chlorophylls are not evident according to ED Figure 6. Moreover, in the original cryo-EM density map shown in ED Figure 6, these chlorophylls do not show clear feature for the oxygen atoms of the 2-formyl group, which should be clear at such a high resolution.

In relation to the Chl f assignment, the authors did not provide any biochemical characterization results (such as the pigment composition analysis), which needs to be done for at least the far-red PSI. I strongly encourage the authors to analyze the pigment composition and quantification for both white PSI and far-red PSI. This should give a clue of how many Chl f molecules present in the far-red PSI sample. And this information can be used in the Chl f assignment in the model building and structure analysis.

Minor point

It is interesting that four subunits are replaced by the homologous gene products in the far-red PSI. The authors could comment more about this and do more comparison. I suggest that the authors compare the sequences of these four subunits in *H. hongdechloris* and in other Chl f-containing cyanobacteria, to show whether these specific sequences are unique and conserved in all Chl f-containing species.

Line 105: all insertions (loop1-4)  all insertions and deletion (loop1-4)

Line 111: Is Chl a865A conserved in PSIs from other species? Please clarify.

Line 126-127: This sentence is confusing. The way of writing can be read as that both white and far-red PSIs contain the PsaK subunit, which binds one additional Chl a in far-red PSI than that in white PSI. However, the PsaK subunit was actually deleted from the white PSI model. This should be clearly stated in the manuscript.

Line 140-141: It seems that the Chl f810 is not quite involved in the Chl f network, since the edge-to-edge distance of 18.0 Å is relatively long for the chlorophyll coupling.

Line 152-153: Again, are these changes conserved in Chl f-containing PSI from other species? It would be better to provide the sequence alignment result.

Line 160-162: Reference is need for this sentence.

Line 196-198: Please delete the figure citation.

Responses to the reviewer's comments

First of all, we reformatted the MS to make it fit with the style of “Nature Communications”, as the previous version was based on a “Nature Report”. The major changes related with the style reformatting include: 1) The number of words in the abstract was reduced following the journal rule, and a part of the previous abstract was moved to Introduction. 2) We added the Discussion section in the revised MS. 3) We added a few more references in the revised manuscript which are not included in the previous version due to space limitations. The content of the MS itself was not changed.

Responses to the Reviewer #1:

(The original comments of the reviewer are depicted in italic)

Comment 1:

Remarks to the Author:

This manuscript addresses important questions of the association of chlorophyll f that absorbs light at the far-red spectrum in PSI that usually utilizes chlorophyll a for light absorption and conversion into chemical energy. The experimental systems were chosen very well and studied by advanced biochemical and structural methods. The main tool was cryo-EM technique the interpretation of which is frequently abused. This is certainly not the case of this manuscript that is well executed, interpreted and written.

Author reply 1:

First of all, we thank the reviewer for his/her highly positive and valuable comments. According to the reviewer's comments, we modified the manuscript as follows.

Comment 2:

*The model building described in the manuscript is highly satisfactory. The authors correctly refrained from including PsaJ, PsaF and PsaK in the model. "The poor densities of PsaF and PsaJ may be due to the lack of the PsaX subunit which is not found in the genome of *H. hongdechloris*, because PsaF and PsaJ are located near PsaX in the PSI-trimer structure and therefore their binding may become weak6." However the explanation is questionable because *Synechocystis* PSI contains no PsaX.*

Author reply 2:

We appreciate the reviewer's important comments. According to the reviewer's comments, we removed the sentences "The poor densities of PsaF and PsaJ may be due to the lack of the PsaX subunit which is not found in the genome of *H. hongdechloris*, because PsaF and PsaJ are located near PsaX in the PSI-trimer structure and therefore their binding may become weak. PsaK may be easily dissociated, as has been reported previously." and added the sentence "The density map for PsaF, PsaJ and PsaK of the

white PSI are rather poor compared with that of the other assigned subunits, and hence, these three subunits were deleted in the structure (Supplementary Figs. 2g and 6). SDS-PAGE analysis of the purified PSI solution sample showed that it contained PsaF, PsaJ and PsaK (Supplementary Fig. 1d and supplementary Table 2). Therefore, the poor densities for these three subunits in the cryo-EM map may be due to the weak association of these subunits with PSI and partial dissociation of them during the grid preparation for cryo-EM. ” in lines 106–113 of the revised MS. We also added a sentence to indicate that the gene for PsaX is not present in the genome of *H. hongdechloris* in lines 102–103 of the revised MS.

Comment 3:

Moreover a novel chlorophyll molecule that is present in the same space complementing a triplet that is a good candidate for a “red trap”. It is advisable to mention red traps and their distribution in the periphery of PSI, in relation to up-hill excitation transfer from chlorophyll f.

Author reply 3:

Thank you very much for the comments.

We added the sentences “The triplet Chls of a836B–a835B–a834B are also found in the *T. elongatus* PSI structure, and they are adjacent to PsaX but not directly related to PsaX⁶. This “red trap” may therefore also work in the *H. hongdechloris* PSI. Another candidate for the red Chl *a* is a839A–a840A³³(Fig. 4b). This component is also conserved between white and far-red PSI.” in line 252–256 in the revised MS.

Comment 4:

They clearly see the [2-formyl]-chlorophyll a and certainly it would not escape detection if present in the ETC where the local resolution is exceptionally high. The chlorophyll f molecules are apparent in the periphery and clearly detected at map densities of higher than 5 sigma. This assessment should be taken as a gold standard for all other cryo-EM structures. The sentence “we found no evidence for participation of chlorophyll f in the ETC” should be included in the abstract not only in the last

sentence of the manuscript.

Author reply 4:

Thank you very much for the encouraging comments. According to the reviewer's comments, we modified the original sentence to "The structure showed that, far-red PSI binds 83 Chl *a* and 7 Chl *f*, and Chl *f* are associated at the periphery of PSI but not in the electron transfer chain." in the Abstract.

Comment 5:

I would highly recommend accepting this manuscript for publication.

Author reply 5:

Thank you very much for your positive comment. We hope that our modifications and responses are satisfactory to you and the Editor.

Responses to the Reviewer #1:

(The original comments of the reviewer are depicted in italic)

Comment 1:

Remarks to the Author:

The manuscript by Kato et al. describes two cryo-EM structures of PSI core from H. hongdechloris grown under white and far-red light conditions. The two structures are solved at overall resolutions of 2.35 and 2.41 Å, respectively. The white PSI contains only Chl a, while the far-red PSI contains Chl f in addition to Chl a, which was induced by the far-red light. Furthermore, the authors observed the replacement of four core subunits with the homologous gene products in the far-red PSI, and identified the role of specific protein subunits and fragments in Chl f binding. The cryo-EM data are sound, and the high resolution map allows the accurate model building. The work would be a great addition to the photosynthesis field, however, I think the study also has some weaknesses as listed below.

Author reply 1:

First of all, we appreciate the reviewer for his/her positive and valuable comments to improve our manuscript. According to the reviewer's comments, we modified the manuscript as follows. We hope that these modifications improved our MS significantly, and will be satisfactory to the reviewer as well as the Editor.

Comment 2:

The first concern is that the authors did not give enough information in the Method section. For example, the description of the PSI purification is very short and does not give any details. The experimental conditions for the measurement of absorption and fluorescence spectra were not described at all. The method section should be described in more details. In addition, I suggest the authors to provide an additional ED figure to show the characterization of both PSI samples, including the SDS-PAGE, the profiles of chromatography and sucrose density gradient centrifugation.

Author reply 2:

Thank you very much for the comments. We added detailed descriptions regarding the purification and characterizations of the samples in the modified Methods section, and also added one Supplementary figure (Supplementary Fig. 1) and one table (Supplementary table 2) to show the purification procedure, SDS-PAGE, the profiles of chromatography and sucrose density gradient centrifugation, and the results of mass/sequence analyses for the assignment of the polypeptide bands.

Comment 3:

It is a big shortcoming that the PSI core is incomplete in both structures. As the authors described in the text, the white PSI core is loosely associated with PsaK, PsaF and PsaJ, which were therefore deleted in the final model. The authors explained the possible reason for the poor densities of PsaF and PsaJ, and attributed it to the lack of the PsaX subunit. However, the explanation is not convinced to this reviewer. The PSI from Synechocystis 6803 does not contain PsaX either, nevertheless, the subunits PsaF and PsaJ are well behaved in the crystal structure (Malavath, BBA, 2018). I believe that the partial loss of the three subunits, especially PsaK (this subunit is present in the far-red PSI), in the white PSI sample is due to the high concentration of β -DDM (0.1%) used for the grid preparation. I'm not sure if the sample was further concentrated before this process (no description in the Method). If so, the β -DDM concentration could be even higher. It is quite common for membrane protein complexes to dissociate and lose some peripheral subunits under conditions of high concentration of detergent. The authors need to explain why they used the higher concentration of β -DDM in grid preparation for the white PSI than that for far-red PSI (0.04%).

Author reply 3:

Thanks for these important comments. We agree with the reviewer's comments that the loss of PsaF, PsaJ may not be due to the absence of PsaX, and therefore removed the following sentences from the revised MS: "The poor densities of PsaF and PsaJ may be due to the lack of the PsaX subunit which is not found in the genome of *H. hongdechloris*, because PsaF and PsaJ are located near PsaX in the PSI-trimer structure

and therefore their binding may become weak. PsaK may be easily dissociated, as has been reported previously.”

It is true that we used a different concentration of β -DDM for the preparation of white and far-red PSI (0.1% vs 0.04%), and the samples were concentrated after the final sucrose density gradient centrifugation by Ultra-amicon (MWCO=100 kDa). However, these concentrations of β -DDM are still low enough to keep these subunits remained in the purified PSI. Also, since we used a membrane with an MWCO of 100 kDa which is larger than the molecular weight of β -DDM micelles to concentrate the samples, we do not consider that the concentration of β -DDM will be significantly higher than what we used so as to induce dissociation of the PSI subunits in question. Indeed, we identified that all PsaK, PsaF and PsaJ are present in the solution samples of both white and far-red PSI. Therefore, we consider that these subunits may be (partially) lost during the cryo-grid preparation. The easier loss of these subunits may be a feature of PSI from cyanobacteria that undergo remodeling of the specific genes upon transition between white and far-red light conditions, which may facilitate the replacement of these subunits between the different light conditions. We modified the original text and added a few sentences in the Discussion section to reflect these situation.

Comment 4:

The far-red PSI contains PsaK, but does not bind PsaF and PsaJ at all. It is not clear if these subunits were lost during purification or grid preparation processes, or they are truly absent in PSI from the beginning. If the subunits PsaF and PsaJ are indeed absent from far-red PSI, the physiological relevance of the loss of PsaF/PsaJ and the adaptation to the far-red light should be discussed. PsaF was previously suggested to be involved in binding electron donor, such as the c-type cytochrome. If the far-red PSI does not contain PsaF, the P700 reduction activity might be affected. The measurement of activity of both white and far-red PSI should be done.

Author reply 4:

Thank you for the critical comments. SDS-PAGE analysis of the purified white and

far-red PSI clearly showed that both white and far-red PSI solution samples contained PsaF, PsaJ and PsaK (Supplementary Fig. 1); thus, we consider that these subunits were lost during cryo-grid preparation as mentioned above. As mentioned above, we consider that the easy loss of these subunits in this cyanobacterial PSI may be a feature for those of PSI from cyanobacteria that undergo remodeling between white and far-red light, which will facilitate easy exchange of these subunits differently expressed the two different light conditions. Indeed, we measured the light-induced difference absorption change of P700 for *Thermosynechococcus elongatus* PSI and far-red PSI, which showed that the difference absorption changes are almost the same between the two PSIs (New Supplementary Fig. 7). This result supports the presence and functioning of PsaF in the purified far-red PSI core solution sample, and is in agreement with the SDS-PAGE analysis. We modified the MS to reflect these situations.

Comment 5:

The authors assigned seven Chl f molecules in each PSI monomer. Based on the cryo-EM densities shown in Extended Data Figure 6, the identity of Chls f826A, f844A, f810B is convincing. However, I have some doubts about the assignment of other four Chl f molecules. Since the difference densities for the 2-formyl groups of these four chlorophylls are not evident according to ED Figure 6. Moreover, in the original cryo-EM density map shown in ED Figure 6, these chlorophylls do not show clear feature for the oxygen atoms of the 2-formyl group, which should be clear at such a high resolution.

Author reply 5:

Thank you for the critical comments. As described in lines 473–481 of the Methods section, the difference map analysis shows two groups of Chl *f* (one is f826A/f844A/f810B and the other one f827A/f830A/f832A/f825B) based on the noise level of the residual densities. At the first group of the Chls *f*-binding sites, some changes of the surrounding amino acids were found which accommodate the binding of Chl *f* (Fig. 3c-e). This resulted in a low noise in the densities because of a high occupancy of Chl *f*. In contrast, at the second group of the Chls *f*-binding sites, no

changes were found in the amino acid residues around the Chl *f*-binding sites (Figs. 3f-g), which resulted in a high noise level in the density because of a lower occupancy of Chl *f*. However, there are counterparts for hydrogen bond to 2-folmyl group of Chls *f* in the second group, and the Chls *f* molecules may be stabilized by these hydrogen bond more than Chl *a*. Li et al. already reported that the amount of Chl *f* in far-red PSI was *ca.* 7.6 (New Ref. 31). The number of Chls *f* we assigned in the present study is almost identical to that study, and the result of our difference map analysis is consistent with the pigment composition analysis. Therefore, we assigned the second group of Chls as Chls *f*.

Comment 6:

In relation to the Chl f assignment, the authors did not provide any biochemical characterization results (such as the pigment composition analysis), which needs to be done for at least the far-red PSI. I strongly encourage the authors to analyze the pigment composition and quantification for both white PSI and far-red PSI. This should give a clue of how many Chl f molecules present in the far-red PSI sample. And this information can be used in the Chl f assignment in the model building and structure analysis.

Author reply 6:

As mentioned above, the Chl *f* content of this cyanobacterium grown under the far-red condition has been measured by Li et al. (Ref. 31), which resulted in 7.6 Chl *f* per PSI monomer. Our structural analysis is very similar to this number of Chl *f* for the far-red PSI. However, we would like to mention that the estimation of the number of Chls in PSI by HPLC analysis may have large deviations. One example for this is the number of Chl *d* in the PSI of *Acaryochloris* estimated by HPLC. The amount of Chl *d* per one Chl *a* was 180 in that paper (Hu et al., Proc. Natl. Acad. Sci. USA, 95, 13319-13323 (1988), DOI: 10.1073/pnas.95.22.13319), which is much larger than other studies. One reason for this large deviation may be that the estimation is based on only one molecule of Chl in PSI. Thus, instead of estimating the number of Chl *f* by HPLC analysis, we deconvoluted the absorption spectrum of far-red PSI at 80K and calculated the areas of

Chl *a* and Chl *f*. The relative areas which considered the molecular coefficients of Chl *a* and Chl *f* were found to be 89.1 and 10.9 for Chl *a* and Chl *f*, respectively, which give rise to a number of Chl *f* of 9.6 for far-red PSI. This number is very close to the number of Chl *f* we assigned in the structure. We added the following sentences to reflect this in the revised MS:

“Other cofactors in the far-red PSI include 83 Chl *a*, 16 β -carotene, 3 [4Fe-4S] cluster, 2 phylloquinones and 2 lipid molecules in a monomer. This gives rise to a Chl *a/f* ratio of 11.8 in the far-red PSI, which is very similar to the results reported recently³¹. The relative areas which considered the molecular coefficients of Chl *a* and Chl *f*² in the 80 K absorption spectrum were 89.1 and 10.9, respectively (Supplementary Fig. 13a, insert). This value is also consistent with the number of Chls *f* found in the structure of the far-red PSI.” in line 191–195 in the revised MS. We also added the sentences in the Discussion section: “Among these Chls, we identified seven Chl *f* in the far-red PSI based on the approach of difference density maps between the cryo-EM maps of white and far-red PSI as well as between the experimental cryo-EM map of the far-red PSI and that calculated with all Chls assigned as Chl *a*. This number agrees well with the ratio of 1:8 for Chl *a/f* found in this cyanobacterium grown under far-red light conditions³¹, and is also comparable with the spectral analysis performed in this study.” in line 309–314 in the revised MS.

Minor point

Comment 7:

It is interesting that four subunits are replaced by the homologous gene products in the far-red PSI. The authors could comment more about this and do more comparison. I suggest that the authors compare the sequences of these four subunits in H. hongdechloris and in other Chl f-containing cyanobacteria, to show whether these specific sequences are unique and conserved in all Chl f-containing species.

Author reply 7:

Thank you for the important comments. According to reviewer’s comments, we compared the sequences of four subunits, PsaA, PsaB, PsaI and PsaL in some species

possessing Chl *f*. Multiple sequence alignment shows that loop2 of PsaA which directly hinders the binding of a865A is conserved in the far-red light induced PSI. The other loops (loop1, 3 and 4) might be specific in *H. hongdechloris*. We added multiple sequence alignment as new Supplementary Figs. 8–11 and Supplementary Table 3, and also added the sentences “Multiple sequence alignment of these four subunits from three species of cyanobacteria that are known to undergo remodeling between white and far-red light conditions (*H. hongdechloris*, *Leptolyngbya* sp. strain JSC-1 and *Chroococcidiopsis thermalis* PCC7203) shows high homologies for each pair of the subunits grown under the same conditions (either white or far-red conditions), but lower homologies between the same subunits expressed under white or far-red conditions (Supplementary Figs. 8–11 and Supplementary Table 3). This is particular apparent for the PsaA and PsaI subunits. For example, the PsaA subunit has an average identity of 84.4% and 83.2% among cyanobacteria grown under far-red and white conditions, respectively, whereas the average identity of PsaA between cyanobacteria grown under far-red and white conditions is 76.2% (Supplementary Table 3). Furthermore, the PsaI subunit has an average identity of 65.0% and 51.8% among cyanobacteria grown under far-red and white conditions, respectively, whereas the average identity between cyanobacteria grown under far-red and white conditions is only 38.6%.” in line 142–156 in the revised MS.

Comment 8:

Line 105: all insertions (loop1-4)  all insertions and deletion (loop1-4)

Author reply 8:

Thank you very much for your suggestion.

We revised the sentence from “all insertions (loop1-4)” to “all insertions and deletion (loop1-4)” in line 165 in the revised MS.

Comment 9:

Line 111: Is Chl a865A conserved in PSIs from other species? Please clarify.

Author reply 9:

Thank you very much for your kind suggestion. Yes, Chl a865 is conserved in PSIs from other species, and we revised the sentences as follows: “The loop2 is conserved in the *psaA* gene expressed under far-red light condition but absent in the gene expressed under white light condition in the Chl *f*-containing species (Supplementary Fig. 8). This loop2 is also absent in PsaA from *Synechocystis* sp. PCC6803 (NCBI:WP_010872067) and *T. elongatus* (NCBI:WP_011056578.1), from which the PSI structure has been solved and the corresponding Chl a685A molecule is present. These results indicate that the far-red light induced expression of different copies of the PSI genes *psaA*, *psaB*, *psaI* and *psaL*, and differences in the sequences of these genes, especially the *psaA* gene, caused the structural changes of the PSI complex, especially with respect to the insertion of the four loop regions in the far-red PSI (Fig. 2 and Supplementary Figs. 8–11).” In line 171–181 in the revised MS.

Comment 10:

Line 126-127: This sentence is confusing. The way of writing can be read as that both white and far-red PSIs contain the PsaK subunit, which binds one additional Chl a in far-red PSI than that in white PSI. However, the PsaK subunit was actually deleted from the white PSI model. This should be clearly stated in the manuscript.

Author reply 10:

Thank you very much for your kind indication. We modified the sentences as follows: “Compared with the white PSI structure, the far-red PSI has one additional Chl *a* (a101K) in the PsaK subunit, as PsaK was lost in the white PSI. However, one Chl *a* (a865A) is depleted in the far-red PSI as described above. As a result, the total number of Chls are the same between the white PSI and far-red PSI.” in line 197–200 in the revised MS.

Comment 11:

Line 140-141: It seems that the Chl f810 is not quite involved in the Chl f network, since the edge-to-edge distance of 18.0 Å is relatively long for the chlorophyll coupling.

Author reply 11:

Thank you very much your comment. We agree that Chl f810 is rather far from the other five Chl *f* molecules. However, considering the coupling between Chl f810B-BRC102I-Chl f844A as well as the similar location of Chl f810B and the five Chls *f* at the same PsaA side, we consider that Chl f810B is also a part of the Chl *f* network. We revised the original sentences to reflect these situations in the revised MS (lines 213–219, lines 261–262, lines 332–333).

Comment 12:

Line 152-153: Again, are these changes conserved in Chl f-containing PSI from other species? It would be better to provide the sequence alignment result.

Author reply 12:

Thank you for the important comment. According to reviewer's comments, we added multiple sequence alignment as new Supplementary Figs. 8–11 and modified the text as mentioned above.

Comment 13:

Line 160-162: Reference is need for this sentence.

Author reply 13:

Thank you for your suggestion.

We added the appropriate references (Refs. 34–36 in the revised MS).

Comment 14:

Line 196-198: Please delete the figure citation.

Author reply 14:

We are sorry for careless mistake. We deleted the figure citation in line 278-279 in the revised MS.

REVIEWERS' COMMENTS:

Reviewer #2 (Remarks to the Author):

The authors have satisfactorily addressed my comments, and the revised manuscript has been greatly improved.